# Purrception: Variational Flow Matching for Vector-Quantized Image Generation

**Răzvan-Andrei Matișan**[†]
UvA-Bosch Delta Lab
University of Amsterdam

**Vincent Tao Hu**
CompVis @ LMU Munich
Munich Center for Machine Learning

**Grigory Bartosh**
AMLab
University of Amsterdam

**Björn Ommer**
CompVis @ LMU Munich
Munich Center for Machine Learning

**Cees G. M. Snoek**
Video & Image Sense Lab
University of Amsterdam

**Max Welling**
UvA-Bosch Delta Lab
University of Amsterdam

**Jan-Willem van de Meent**
UvA-Bosch Delta Lab
University of Amsterdam

**Mohammad Mahdi Derakhshani**[*]
Video & Image Sense Lab
University of Amsterdam

**Floor Eijkelboom**[*]
UvA-Bosch Delta Lab
University of Amsterdam

## Abstract

We introduce Purrception, a variational flow matching approach for vector-quantized image generation that provides explicit categorical supervision while maintaining continuous transport dynamics. Our method adapts Variational Flow Matching to vector-quantized latents by learning categorical posteriors over codebook indices while computing velocity fields in the continuous embedding space. This combines the geometric awareness of continuous methods with the discrete supervision of categorical approaches, enabling uncertainty quantification over plausible codes and temperature-controlled generation. We evaluate Purrception on ImageNet-1k $256 \times 256$ generation. Training converges faster than both continuous flow matching and discrete flow matching baselines while achieving competitive FID scores with state-of-the-art models. This demonstrates that Variational Flow Matching can effectively bridge continuous transport and discrete supervision for improved training efficiency in image generation.

## 1 Introduction

The task of generative modeling is to approximate a data distribution to enable sampling of new instances. Beyond high-fidelity synthesis in images, audio, and text, generative models are increasingly used for augmentation, restoration, simulation, and in-silico design (e.g., de novo molecules and proteins). Flow Matching (Lipman et al., 2023; Albergo et al., 2023; Liu et al., 2023) has emerged as an extremely effective approach for the generation of a variety of data modalities. In Flow Matching, one first defines an interpolation between a source (noise) and a target (data) distribution, and then approximate the velocity field of a continuous normalizing flow that transports samples between the two. As the target velocity can be understood as the expected time-derivative of the interpolation, it can be learned in a self-supervised manner by averaging over samples from the source and target distribution. The Flow Matching framework has been extended to general geometries (Chen & Lipman, 2024), discrete data (Gat et al., 2024), and has seen many applications (Wildberger et al., 2024; Dao et al., 2023; Hu et al., 2024; Kohler et al., 2023).

Variational Flow Matching (VFM) (Eijkelboom et al., 2024) reframes Flow Matching as inference. Since the Flow Matching velocity field is the expectation of a conditional velocity, it can be approximated via a variational posterior over endpoints (target samples) given the current interpolation point. Standard Flow Matching is recovered when this posterior is Gaussian, while other choices extend naturally to different modalities. Applied to discrete data, VFM yields *CatFlow*, previously used

---

[†]Research done during an internship at UvA-Bosch Delta Lab and a visit at LMU Munich.
[*]Equal contribution as last authors.

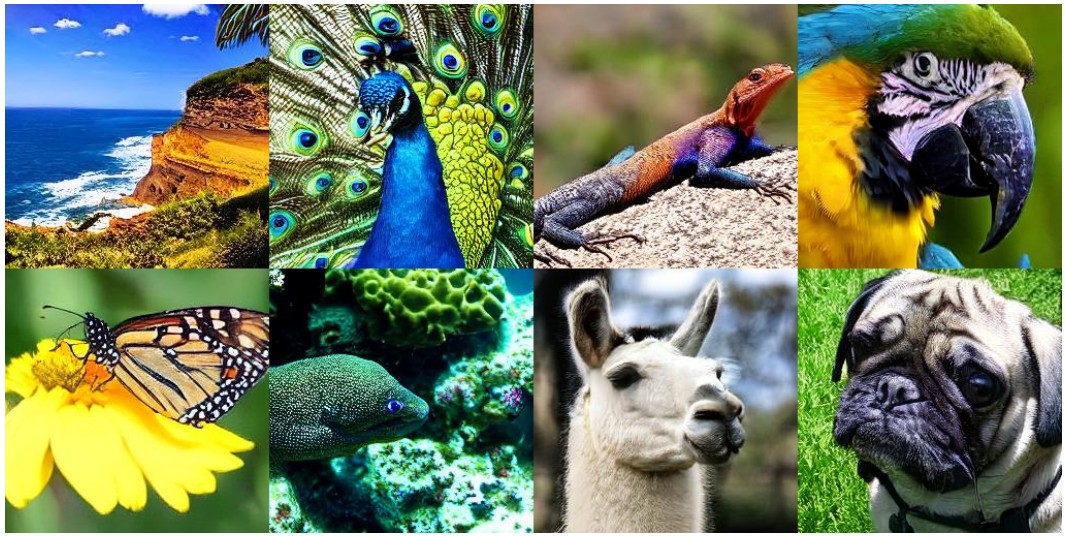

Figure 1: Purrception generates high-resolution images in vector-quantized latent spaces, sampled as continuous transport learned through discrete supervision.

for graph generation and related to continuous diffusion for categorical data (Dieleman et al., 2022). More broadly, VFM has been applied to mixed modalities (Guzmán-Cordero et al., 2025), molecular generation (Eijkelboom et al., 2025; Sakalyan et al.), and general geometries (Zaghen et al., 2025). The variational view also enables problem-specific constraints, e.g., for sea-ice forecasting, where bounds like non-negative thickness are enforced through the loss (Finn et al., 2025).

This paper leverages VFM in the context of image generation. We consider vector-quantized (VQ) latents, which map images into grids of discrete indices with associated embeddings, yielding compact representations that preserve perceptual fidelity at far lower dimensionality than pixels. However, their dual discrete–continuous nature poses a modeling challenge not addressed by purely continuous or discrete methods. Continuous methods (latent diffusion, flow matching) generate in the embedding space, enabling smooth transport and efficient high-resolution synthesis (Rombach et al., 2022; Dao et al., 2023). Yet they must discretize vectors back to indices: geometry is preserved, but categorical structure is ignored – the model never learns which index to choose or how to represent uncertainty across codes. Conversely, fully discrete approaches (VQ-Diffusion (Gu et al., 2022), discrete flow matching (DFM) (Gat et al., 2024)) treat related embeddings as unrelated tokens, discarding geometry. While DFM could use temperature-based sampling, this only produces stochastic "hops" between indices – each step commits to a single code – whereas continuous flow matching (CFM) cannot use temperature at all, since it lacks logits.

To resolve this tradeoff, we introduce Purrception, an adaptation of VFM to vector-quantized latents. By using a categorical posterior over indices while transporting probability in the continuous embedding space, Purrception provides a categorical learning signal while still leveraging geometry. This means the model can express uncertainty across plausible codes and translate it into smooth, geometry-aware transport rather than discrete jumps. Logits further enable temperature scaling: lowering temperature sharpens predictions, while raising temperature spreads probability across nearby embeddings, producing smoother generations and samples with more details. Empirically, this hybrid approach converges faster than both CFM and DFM on ImageNet-1k, achieving competitive or superior FID while retaining the efficiency of flow matching.

## 2 BACKGROUND

### 2.1 FLOW MATCHING

Flow matching (Lipman et al., 2023; Liu et al., 2023; Albergo et al., 2023) learns a velocity field $v_t^\theta : \mathbb{R}^D \times [0, 1] \to \mathbb{R}^D$ – parameterized by a network with parameters $\theta$ – which induces a transport

of samples $x_0 \sim p_0$ from a prior (e.g., standard noise) to $D$-dimensional points $x_1$ that should approximate the data distribution. This is done by integrating the ordinary differential equation

$$\frac{\mathrm{d}x}{\mathrm{d}t} = v_t^\theta(x) \qquad \text{with } x_0 \sim p_0, \qquad (1)$$

which is equivalent to learning a velocity field that satisfies the continuity equation, also known as a continuous normalizing flow,

$$\partial_t p_t(x) = -\nabla \cdot \left( v_t^\theta(x) p_t(x) \right). \qquad (2)$$

Flow matching starts from the observation that, given a choice of interpolation between noise and data – e.g., linear, where $x_t = tx_1 + (1-t)x_0$ – we can derive a conditional velocity field $u_t(x \mid x_1)$ that satisfies the continuity equation towards (i.e., conditional on) a specific endpoint. A corresponding velocity field $u_t(x)$, which satisfies the continuity equation for the (marginal) probability path, can be expressed in terms of an (intractable) expectation with respect to the posterior

$$u_t(x) = \int u_t(x \mid x_1) \, p_t(x_1 \mid x) \, \mathrm{d}x_1 = \mathbb{E}_{p_t(x_1 \mid x)} \left[ u_t(x \mid x_1) \right]. \qquad (3)$$

The goal of flow matching is therefore to learn a velocity field $v_t^\theta(x)$ that approximates $u_t(x)$, i.e., to minimize the flow matching objective

$$\mathcal{L}_{\text{FM}}(\theta) = \mathbb{E}_{t,x} \left[ ||v_t^\theta(x) - u_t(x)||^2 \right], \qquad (4)$$

which can be made tractable by optimizing

$$\mathcal{L}_{\text{CFM}}(\theta) = \mathbb{E}_{t,x_1,x} \left[ ||v_t^\theta(x) - u_t(x \mid x_1)||^2 \right], \qquad (5)$$

i.e., a Monte-Carlo estimate of the marginal objective through our conditional objective. As shown in Lipman et al. (2023), indeed these two objectives have the same gradients w.r.t. $\theta$. This can equivalently be understood as trying to regress towards the expected time-derivative of the interpolant.

## 2.2 VARIATIONAL FLOW MATCHING

Variational Flow Matching (VFM) (Eijkelboom et al., 2024) treats Flow Matching as a variational inference problem. By realizing (through Equation (3)) the target marginal velocity field $u_t$ can be expressed as an expectation of the conditional field w.r.t. the posterior distribution $p_t(x_1 \mid x)$, the authors propose to learn this posterior directly, i.e., learn

$$\mathcal{L}_{\text{VFM}}(\theta) := \mathbb{E}_t \left[ \text{KL}(p_t(x_1, x) \mid\mid q_t^\theta(x_1, x)) \right] = -\mathbb{E}_{t,x_1,x}[\log q_t^\theta(x_1 \mid x)] + \text{const.}, \qquad (6)$$

where $q_t^\theta(x_1 \mid x)$ is the *variational posterior* approximating the posterior probability path $p_t(x_1 \mid x)$. The resulting learning velocity field is thus given by

$$v_t^\theta(x) := \mathbb{E}_{q_t^\theta(x_1 \mid x)} \left[ u_t(x \mid x_1) \right] \overset{\text{OT}}{=} \frac{\mu_t^\theta(x) - x}{1-t}, \qquad (7)$$

where $\mu_t^\theta(x) := \mathbb{E}_{q_{t}^\theta}[x_1 \mid x]$ and the conditional field is the linear (or optimal transport) interpolation. Though this objective initially looks intractable, we authors show that the task of learning the variational approximation only needs to be learned dimension-wise in the mean, as $\mathbb{E}_{q_t^\theta(x_1 \mid x)}[x_1^d \mid x]$ only depends on the marginal $q_t^\theta(x^d \mid x)$ – an approach called *mean-field VFM*.

VFM is flexible in choosing the variational distribution $q_t^\theta$, which makes it a general framework for different data types. In Eijkelboom et al. (2024), the authors show significant improvement over CFM when the data is discrete and the variational approximation is chosen to be categorical, a model coined *CatFlow*. VFM has also obtained strong performance in tabular data (Guzmán-Cordero et al., 2025), molecular generation tasks (Eijkelboom et al., 2025; Sakalyan et al.), general geometries (Zaghen et al., 2025), and sea-ice modeling (Finn et al., 2025).

## 2.3 VECTOR-QUANTIZED AND LATENT GENERATIVE MODELS

High-resolution image modeling in pixel space is computationally prohibitive; a common remedy is to learn a lower-dimensional latent space with an autoencoder. Vector-Quantized VAEs (Van Den Oord

et al., 2017) and VQ-GANs (Esser et al., 2021) use a discrete codebook $\mathcal{C} = \{e_k\}_{k=1}^{K} \subseteq \mathbb{R}^D$. By mapping images into a compact set of discrete tokens, vector-quantized latents provide an efficient and stable representation: they alleviate posterior collapse and often yield sharper, higher-fidelity reconstructions than pixel-space models at comparable compute.

Given an image $x$, the encoder output is *quantized* to its nearest code:

$$z(x) \;=\; \text{Quantize}(\text{Encoder}(x)) \;=\; \underset{e_k \in \mathcal{C}}{\arg\min} \left\|\text{Encoder}(x) - e_k\right\|_2^2. \tag{8}$$

Equivalently, one can store the index

$$c(x) = \underset{k \in [K]}{\arg\min} \left\|\text{Encoder}(x) - e_k\right\|_2^2, \qquad [K] := \{1, \ldots, K\}. \tag{9}$$

After training the encoder, decoder, and codebook, a generative model is learned *in latent space* and samples are decoded to pixels. For a grid of $D$ discrete latents $c \in [K]^D$, a common choice is an autoregressive model:

$$p(c) = \prod_{d=1}^{D} p\left(c_d \mid c_{<d}\right). \tag{10}$$

While this formulation provides a powerful and efficient representation, it also introduces a fundamental modeling tension: each latent is at once a discrete code index and a continuous embedding vector. Existing generative methods resolve this tension by making a trade-off – either operating in the continuous embedding space and ignoring the categorical structure, or modeling indices directly while discarding geometric information. This limitation motivates the hybrid perspective developed in Section 3.

## 3 PURRCEPTION: VQ-VFM FOR IMAGES

### 3.1 MOTIVATION: A HYBRID APPROACH TO VQ-LATENT FLOWS

Vector-quantized (VQ) latents encode data in two ways simultaneously: as discrete indices drawn from a finite codebook and as continuous embeddings that capture geometric relations such as proximity and direction. Existing generative models are typically forced into one of two degenerate extremes, each of which breaks part of this dual structure:

- **Continuous flow models** (e.g., latent diffusion and flow matching) operate in $\mathbb{R}^D$, treating codebook vectors as continuous. From the perspective of Variational Flow Matching (VFM), this corresponds to a Gaussian relaxation: endpoints are approximated as continuous samples rather than categorical indices. Geometry is preserved, but discreteness is lost – the model never receives a categorical learning signal, cannot express uncertainty over multiple plausible codes, and has no logits from which to derive controls such as temperature scaling.

- **Fully discrete flow models** instead predict categorical indices directly. While this aligns with the quantized structure, it collapses geometry: once reduced to raw indices, semantically related codes are treated as unrelated tokens. Predictions degenerate into discrete "teleports" between indices, eliminating interpolation and making both uncertainty modeling and temperature scaling meaningless.

An ideal solution should combine the strengths of both worlds: exploit the smooth geometry of embeddings *and* provide categorical supervision over indices. Our approach adapts VFM with a *categorical variational posterior*, so that the velocity field evolves in continuous space while learning is driven by cross-entropy over codebook entries. This hybridization allows the model to receive a categorical learning signal, to reason over multiple plausible indices, and to convert that uncertainty into geometry-aware transport rather than discrete jumps. Crucially, working with logits also unlocks a temperature knob: lowering $\tau$ enforces stronger commitments, which improves global fidelity but oversimplifies samples, while raising $\tau$ redistributes probability more broadly, adding detail and variety at the cost of overall quality.

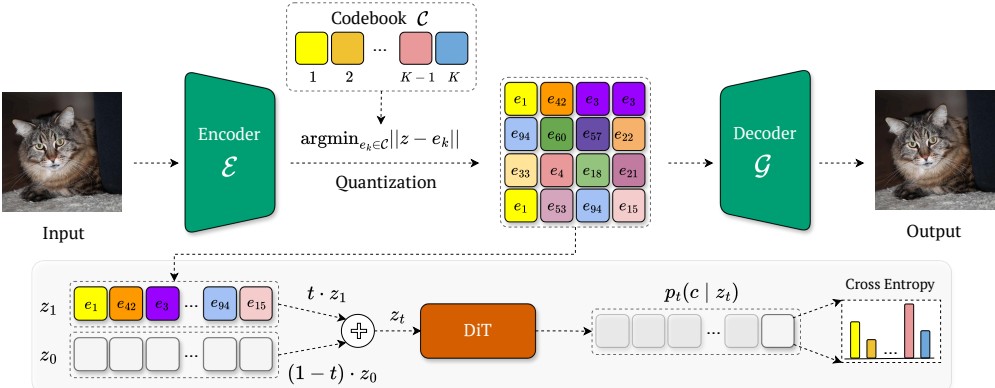

Figure 2: **Purrception approach.** Purrception generates high-resolution images in a vector quantized latent space. For training, we use a pretrained encoder $\mathcal{E}$ and a codebook vector of size $K$ to encode and quantize an image in latent space to obtain $z_1$. Then, we train a diffusion transformer that predicts, given a linear interpolant $z_t$, a categorical distribution over the codebook vectors for each patch of the target $z_1$ via a cross-entropy objective. For sampling, we generate a quantized latent which we further pass through the decoder $\mathcal{G}$ to obtain the image in pixel-space.

## 3.2 THE VQ-VFM OBJECTIVE

We begin from the key observation underlying VFM and CatFlow: the velocity at time $t$ can be expressed as an expectation over conditional continuations weighted by a posterior over endpoints (Eijkelboom et al., 2024):

$$u_t(z_t) = \mathbb{E}_{p_t(z_1|z_t)}\big[u_t(z_t \mid z_1)\big]. \tag{11}$$

This perspective reframes the learning problem: rather than predicting the vector field directly, we may approximate the posterior $p_t(z_1 \mid z_t)$ with a variational distribution $q_t^\theta(z_1 \mid z_t)$ and compute the velocity as its expectation.

In the case of VQ-latents, this insight becomes particularly powerful. Each endpoint $z_1$ must be one of the finite codebook embeddings $\{e_k\}_{k=1}^K$, so the posterior *is* categorical over the discrete latent codes. That is, our variational posterior should be given by

$$q_t^\theta(c \mid z_t) = \mathrm{Cat}(c \mid \pi_t^\theta(z_t)). \tag{12}$$

where $\pi_t^\theta(z_t)$ is the probability distribution over the codebook vectors outputted by a neural network (e.g., Diffusion Transformer (Peebles & Xie, 2023)). Conditioning this posterior on the noisy latent $z_t$ yields a distribution over discrete indices while still defining transport in continuous embedding space, as we can compute

$$v_t^\theta(z_t) = \sum_{k=1}^K \pi_t^{\theta,k}(z_t)\left(\frac{e_k - z_t}{1-t}\right) = \frac{\mu_t(z_t) - z_t}{1-t}, \tag{13}$$

where $\mu_t(z_t) := \sum_{k=1}^K \pi_t^{\theta,k}(z_t)e_k$ and $\pi_t^{\theta,k}(z_t)$ is the probability to have as endpoint the codebook vector $e_k$ given the time-dependent interpolant $z_t$. This ensures that uncertainty over multiple plausible codes is translated into smooth, geometry-aware motion, rather than discrete "teleports" between unrelated indices.

Training follows from the VFM objective, which in this case reduces to the cross-entropy loss between the predicted posterior and the ground-truth code indices:

$$\mathcal{L}_{\mathrm{Purr}}(\theta) = -\mathbb{E}_{t,x,z_t}\big[\log q_\theta(c \mid z_t)\big], \tag{14}$$

where $x \sim \mathcal{D}$ is sampled from the data, $z_1$ and $c$ is the corresponding quantized image and latent code respectively, and $z_t$ is obtained through $z_t := tz_1 + (1-t)z_0$ for $z_0 \sim p_0$ and $t \sim \mathcal{U}(0,1)$.

**Softmax temperature.**  Because $\pi_t^\theta(z_t)$ is obtained from logits $\tilde{\pi}_t^\theta(z_t)$ via a softmax with temperature $\tau$,

$$\pi_t^{\theta,k}(z_t) = \frac{\exp(\tilde{\pi}_t^{\theta,k}(z_t)/\tau)}{\sum_{i=1}^K \exp(\tilde{\pi}_t^{\theta,i}(z_t)/\tau)}, \tag{15}$$

our framework naturally inherits an inference-time degree of freedom that regulates how categorical uncertainty is expressed in the velocity field. When $\tau$ is small, the posterior distribution collapses toward the most likely index, enforcing early commitments and producing sharp, high-fidelity outputs that may, however, become overly simplistic as alternative hypotheses are ignored. Conversely, large $\tau$ values flatten the distribution, assigning non-negligible weight to multiple neighboring codes. This broadening injects more detail and variability into the generated samples, but can reduce overall fidelity as the barycenter drifts away from the most plausible embedding. Intermediate $\tau$ values often strike the best balance, echoing the bias–variance trade-off familiar from other generative frameworks. Such controllability is absent in continuous FM, where no categorical logits exist, and meaningless in fully discrete FM, where indices are collapsed immediately; it arises directly from the hybrid VQ–VFM formulation, turning temperature into a principled knob for task-adaptive inference.

## 4 EXPERIMENTS AND RESULTS

We validate the performance of Purrception through a series of experiments. In our experiments, we evaluate on ImageNet-1k (Deng et al., 2009) on $256\times256$ resolution, using both the Stable Diffusion's `vq-f8` (Esser et al., 2021) and LlamaGen's `vq-ds8-c2i` (Sun et al., 2024) tokenizers, as well as the DiT-L/2 and DiT-XL/2 backbones (Peebles & Xie, 2023). We provide a full description of the implementation details in Appendix C. First, we perform a comparative study between Purrception, continuous flow matching (Lipman et al., 2023), and discrete flow matching (DFM) (Gat et al., 2024). For continuous flow matching, we consider two objectives: the classical regression task of predicting the velocity field (denoted simply as CFM) and the task of predicting the endpoint (denoted as CFM-endpoint), as seen in Ma et al. (2024), allowing us to measure the effects of both (1) switching to endpoint prediction, and (2) using our discrete objective compared to the continuous baseline. We show that Purrception *converges faster* (i.e., in fewer training iterations) to a low FID, hence reducing computational resources. Then, we show that Purrception generates high-fidelity and high-quality samples when trained at scale, achieving a competitive FID against a variety of state-of-the-art autoregressive, diffusion, and masked generative baselines. Finally, we show that the softmax temperature parameter can be used to *control* the image sharpness and quality at inference time, a property unique to hybrid discrete-continuous models.

### 4.1 CONVERGENCE SPEED

A key requirement for practical generative modeling is the ability to reach high sample quality quickly, since faster convergence directly reduces training cost and compute requirements. To evaluate this, we compare the convergence speed of Purrception against two strong baselines: the continuous flow model (CFM) (Lipman et al., 2023) and the fully discrete flow model (DFM) (Gat et al., 2024). Given the great performance of Scalable Interpolant Transformers (SiT) Ma et al. (2024), we include an additional baseline (denoted as CFM-endpoint) where the task is to predict via mean-squared error (similar to CFM) the endpoint $z_1$ given the interpolant $z_t$ (similar to Purrception). For a fair comparison, we used the same training configurations, and we sample all images using Euler with 100 integration steps as ODE solver. We provide all implementation details in Appendix C.

Figure 3 reports FID-10k scores for both DiT-L/2 and DiT-XL/2 backbones. Across settings, Purrception not only achieves lower final FID but also reaches baseline performance substantially earlier. With DiT-L/2, Purrception checkpoint at 2M iterations matches CFM's and CFM-endpoint's scores after ∼1.2M iterations (1.65× faster), while Purrception checkpoint at 1M iterations matches DFM's final score after ∼325k iterations (3.0× faster). With the larger DiT-XL/2 backbone, the gap grows further: Purrception converges 2.3× faster than both CFM baselines and 3.5× faster than DFM.

These results underscore the advantage of Purrception's hybrid formulation. By receiving direct categorical supervision (unlike CFM), the model learns discrete structure more efficiently, while its use of continuous embedding space (unlike DFM) enables smooth geometry-aware transport

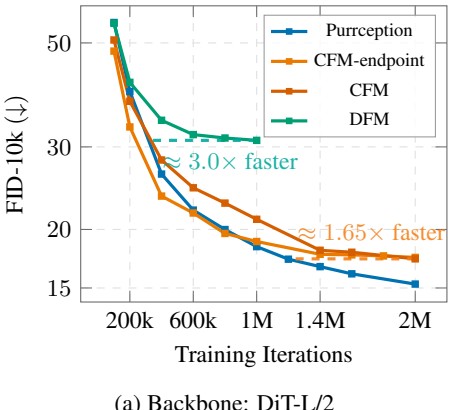
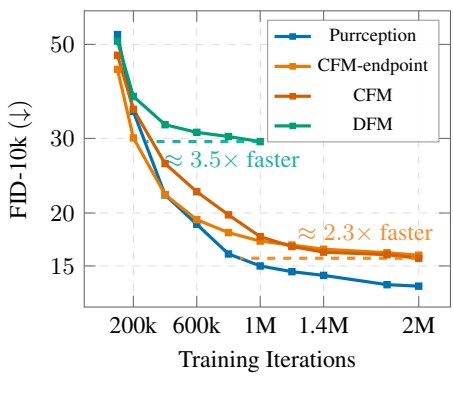

(a) Backbone: DiT-L/2    (b) Backbone: DiT-XL/2

Figure 3: **Convergence speed comparison on ImageNet-1k.** FID-10k scores are plotted against training iterations for Purrception, two CFM variants, and DFM. Results are shown for two DiT backbones: (a) DiT-L/2 and (b) DiT-XL/2. We train Purrception using the default $\tau = 1.0$ softmax temperature, while using $\tau = 0.9$ during inference. The plots show that Purrception achieves lower final FID scores and converges significantly faster, matching the final performance of CFM and DFM in fewer training iterations. Here we used Stable Diffusion's `vq-f8` tokenizer. Full training details are provided in Appendix C.

rather than slow, discrete jumps. This combination accelerates optimization, leading to both faster convergence and stronger sample quality.

## 4.2 Optimizing Sample Quality via Softmax Temperature Scaling

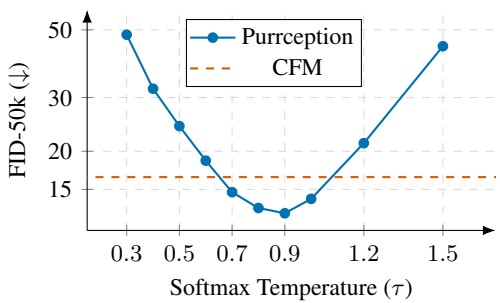

Figure 4: **The effect of the softmax temperature on FID score.** The plot depicts a U-shape relationship between $\tau$ parameter used in Purrception and the FID score. Both models have been trained for 1M iterations and under the same training conditions. Here we used the `vq-f8` autoencoder.

Temperature scaling is a long-standing technique in language modeling, used to balance coherence and diversity during sampling. In the context of VQ image synthesis, continuous flow methods (e.g., CFM) cannot exploit this mechanism at all, since they lack categorical logits. Fully discrete models (e.g., DFM) can in principle apply temperature scaling to their logits, but because they commit to hard index selections at each step, adjusting $\tau$ has little practical effect – the sampling collapses to discrete jumps regardless of the distribution's softness. *In contrast, Purrception retains uncertainty in the logits while transporting through the continuous embedding space, which means temperature scaling can be naturally used.*

To test the effect of the softmax temperature during inference, we conduct an ablation study with a DiT-XL/2 backbone trained for one million iterations. During training, we keep $\tau$ to the default 1.0, varying the temperature *only* at inference. Figures 4 and 5 show the FID-50k obtained and the effect on sample quality, respectively. We observe a clear U-shaped curve: performance improves as $\tau$ increases from very low values, reaches an optimum around $\tau \approx 0.8$–$0.9$, and then degrades as $\tau$ becomes larger. Qualitatively, low $\tau$ values produce overly deterministic and simplistic images, while high $\tau$ values lead to noisy and incoherent generations.

These findings highlight that: (1) even though Purrception has been trained with a constant $\tau = 1.0$, the data distribution is best approximated for lower softmax temperatures, and (2) adjusting $\tau$ is a simple, training-free approach to improve the sample quality. Future work could consist of developing principled softmax temperature schedules during inference or varying $\tau$ during training.

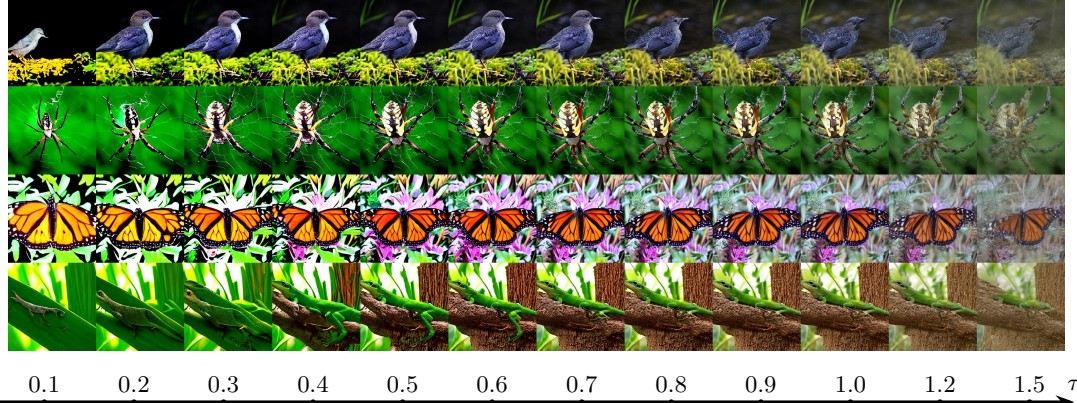

Figure 5: **Generated samples at different softmax temperatures.** We can control the output of Purrception by changing the softmax temperature. A low temperature creates simpler, cleaner samples, while a high temperature adds more fine-grained details but can sometimes introduce flaws and reduce the image quality. Here we vary $\tau$ from 0.1 to 1.5.

### 4.3 QUALITATIVE AND QUANTITATIVE RESULTS

To test how well Purrception performs against similar methods, we train Purrception at scale for 3.5M iterations with a DiT-XL/2 backbone, and report quantitative results on class-conditional ImageNet-1k generation at $256 \times 256$ resolution.

Table 1 highlights a comparison with popular image generation methods, similar to Purrception in model size and methodology, including autoregressive methods (Esser et al., 2021; Yu et al., 2021; Lee et al., 2022; Sun et al., 2024), discrete diffusion and masked generative models (Chang et al., 2022; Gu et al., 2022; Hu & Ommer, 2024), as well as continuous diffusion models (Dhariwal & Nichol, 2021; Ho et al., 2022; Rombach et al., 2022; Peebles & Xie, 2023; Ma et al., 2024). Purrception is competitive in FID score. Notably, Purrception outperforms all discrete diffusion and masked generative models. It also shows stronger performance against most autoregressive methods while having less parameters and/or benefiting from natively faster decoding than large-token autoregressive models (which often rely on inference optimizers such as vLLM Sun et al. (2024)). This firmly establishes Purrception as a novel, state-of-the-art approach, among VQ-based latent generative models, demonstrating that our hybrid discrete-continuous formulation can surpass traditional VQ approaches in fidelity.

Against strong continuous diffusion baselines, Purrception falls short on important baselines like DiT-XL/2 and SiT-XL/2 baselines. We believe this is mainly due to two reasons: (1) the use of high-quality VAE autoencoders in those models, which are known to produce lower FID scores than VQ tokenizers at equivalent scales, and (2) their considerably longer training schedules (twice as many iterations as used for Purrception). Despite this gap, Purrception's strong results highlight that our hybrid design can approach the performance of top-tier diffusion models. This underscores that Purrception effectively bridges the fidelity of continuous diffusion with the categorical training objective suitable for VQ latent spaces, positioning it as a promising direction for future generative modeling.

## 5 RELATED WORK

**Diffusion, flow matching, and latent spaces.** Diffusion and score-based models synthesize data via iterative denoising (Sohl-Dickstein et al., 2015; Ho et al., 2020; Song et al., 2020), while Flow Matching learns a time-dependent velocity field that transports a source distribution to the data distribution, yielding continuous normalizing flows with strong empirical results (Lipman et al., 2023; Liu et al., 2023; Albergo et al., 2023). Moreover, alternative parameterizations of flow matching exist, e.g. endpoint prediction shows improved performance in tasks like image generation and molecular generation (Ma et al., 2024; Eijkelboom et al., 2024). To reduce cost without sacrificing quality, many

Table 1: **Class-conditional generation on ImageNet-1k** $256 \times 256$. We compare Purrception against various autoregressive, diffusion, and masked generative models. We report the number of parameters in millions (M) or billions (B), as well as the FID scores for each model. Purrception achieves a competitive FID of 3.88, showcasing the effectiveness of our hybrid discrete-continuous approach against strong baselines, particularly the VQ image generation methods. Here we use the LlamaGen's `vq-ds8-c2i` (Sun et al., 2024) tokenizer and Euler with 250 integration steps as ODE solver for FID computation.

| Model | #Parameters | FID $\downarrow$ |
|---|---|---|
| *Autoregressive & Masked Generative Models* | | |
| VQGAN (Esser et al., 2021) | 1.4B | 5.20 |
| ViT-VQGAN (Yu et al., 2021) | 1.7B | 3.04 |
| RQTransformer (Lee et al., 2022) | 3.8B | 3.80 |
| LlamaGen-XL (Sun et al., 2024) | 775M | 3.39 |
| MaskGIT (Chang et al., 2022) | 227M | 6.18 |
| Open-MAGVIT2-L (Luo et al., 2024) | 804M | 2.51 |
| *Continuous Diffusion* | | |
| ADM (Dhariwal & Nichol, 2021) | 554M | 10.94 |
| CDM (Ho et al., 2022) | - | 4.88 |
| LDM-4 Rombach et al. (2022) | 400M | 3.60 |
| DiT-XL/2 (Peebles & Xie, 2023) | 675M | 2.27 |
| SiT-XL/2 (Ma et al., 2024) | 675M | 2.06 |
| *Discrete Diffusion & Masked Generative Models* | | |
| VQ-Diffusion (Gu et al., 2022) | - | 5.84 |
| Implicit Timestep Model Hu & Ommer (2024) | 546M | 5.30 |
| ***Hybrid Discrete-Continuous Models*** | | |
| **Purrception** ($\tau = 0.9, \mathrm{cfg} = 1.3$) | **750M** | **3.88** |

works apply these dynamics in vector-quantized latent spaces, where autoencoders provide compact discrete indices with associated embeddings (Van Den Oord et al., 2017; Razavi et al., 2019). Such latents underlie VQ-GAN and large-scale generative systems (Esser et al., 2021; Ramesh et al., 2021; 2022), and running diffusion/flows on them enables efficient high-fidelity synthesis (Vahdat et al., 2021; Rombach et al., 2022; Dao et al., 2023), with recent work scaling to stronger backbones and resolutions (Ma et al., 2024; Esser et al.).

**Discrete dynamics and relaxations.** Beyond continuous latents, discrete diffusion and flow models operate directly on tokens or pixels (Hoogeboom et al., 2021a;b; Austin et al., 2021; Gat et al., 2024; Stark et al., 2024; Davis et al., 2024). Closer to our setting, discrete *latent* diffusion denoises over VQ indices (Gu et al., 2022; Tang et al., 2022), making the indices explicit but typically discarding the geometry of their embeddings. A complementary approach is to embed categorical data into a continuous space and run diffusion there, as in Continuous Diffusion for Categorical Data (CDCD) (Dieleman et al., 2022), developed primarily for language modelling. CDCD preserves the continuous-time formulation by operating on noisy embeddings while training with cross-entropy over token predictions, thereby capturing uncertainty and retaining guidance mechanisms. However, because the embeddings are learned jointly, the approach relies on continuous relaxations and may diverge from the true categorical structure. Our approach follows the same general spirit of combining categorical supervision with continuous transport.

## 6 CONCLUSIONS

We introduced Purrception, an adaptation of VFM to vector-quantized image generation. The method retains continuous transport in the embedding space while supervising with a categorical posterior over codebook indices. This coupling addresses the core trade-off of existing approaches: unlike CFM, Purrception benefits from categorical supervision, and unlike DFM, it avoids collapsing geometry into hard index jumps. The result is a model that learns, broadly speaking, what to choose and where to go, expressing uncertainty over plausible codes in a geometry-aware way. Empirically, Purrception outperforms both CFM and DFM on ImageNet-1k $256 \times 256$ benchmark, converging

faster and achieving superior FID while preserving the efficiency of flow matching. Further ablations confirm that logits provide a controllable quality–diversity knob through temperature scaling.

**Limitations and Future Work.** Our approach is currently limited by its reliance on a fixed, pretrained VQ autoencoder, which makes performance dependent on the initial tokenization quality. While the model is competitive on $256 \times 256$ ImageNet-1k, its generalization to other datasets or higher resolutions needs validation, and it does not yet match the performance of top-tier continuous diffusion models. Future work could directly address these limitations by exploring different VQ models or jointly training the autoencoder with the flow model. Broader research directions include extending this hybrid perspective to domains like audio, video, and 3D shapes, as well as developing principled temperature schedules and a stronger theory for categorical objectives. Finally, because the model remains a continuous flow, it supports distillation into highly efficient, few-step samplers and can incorporate guidance, paving the way for practical generative pipelines.

**Ethics Statement.** All experiments in this work rely exclusively on *publicly available* datasets (i.e., ImageNet) used under their original licenses. We do not collect or annotate any new human data. As with other generative models, there exists a risk of misuse in privacy-invasive or unauthorized applications. We strongly caution against such uses and emphasize the importance of adhering to license terms, governance standards, and applicable legal requirements, though, as our approach is primarily methodological, we do not see immediate risks.

**Reproducibility Statement.** We provide pseudocode for training and sampling Purrception (Appendix B) as well as detailed implementation specifics in Appendix C, which covers optimization settings and evaluation protocols. To facilitate replication, we release the full codebase[*].

**Acknowledgements.** This project was supported by the Bosch Center for Artificial Intelligence. It was also made possible in part by the ELLIS unit in Amsterdam, whose funding supported a collaborative research visit to LMU Munich.

---

[*]https://github.com/razvanmatisan/purrception

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

## A    USAGE OF LARGE LANGUAGE MODELS

During the preparation of this submission, Large Language Models (LLMs) were utilized as a tool to enhance the quality and presentation of our work. Specifically, we employed LLMs for text refinement, including improving grammar, syntax, and clarity to ensure the readability of our research. Additionally, these models assisted in refining the aesthetic and structural layout of our data visualizations and plots, providing suggestions for more effective data presentation. It is important to note that the LLMs served solely as an assistive tool. The authors retained full responsibility for all intellectual content, including the underlying research, data analysis, interpretation of results, and the final articulation of all arguments and conclusions presented in this paper.

## B    ALGORITHMS

TRAINING

**for** $x \sim \mathcal{D}$ **do**
    $z_1 \leftarrow$ Quantize(Encoder($x$));
    $c \leftarrow$ LatentCode($z_1$);
    $z_0 \sim p_0$;
    $t \sim \mathcal{U}(0,1)$ ;
    $z_t \leftarrow tz_1 + (1-t)z_0$;
    $\mathcal{L}(\theta) =$ CrossEntropy($c, \pi_t^\theta(z_t)$);
    Backprop and update $\theta$;
**end**

SAMPLING (EULER INTEGRATION)

$z_0 \sim p_0$;
**for** $s \in \{0, \cdots, T-1\}$ **do**
    $t \leftarrow s/T$;
    $\pi_t \leftarrow$ softmax($\tilde{\pi}_t^\theta(z_s), \tau$);
    $v_s \leftarrow \dfrac{\sum_{k=1}^{K} \pi_t^k \cdot e_k - z_s}{1-t}$;
    $z_{s+1} \leftarrow z_s + (1/T)v_s$;
**end**
$x \leftarrow$ Decoder(Quantize($z_T$));
Return $x$;

Figure 6: Training and sampling algorithms for Purrception.

## C    IMPLEMENTATION DETAILS

**Training specifications.** We use DiT architectures of different sizes as backbones for all models (i.e., Purrception, CFM, DFM). To train them, we mostly use the specifications from the original paper (Peebles & Xie, 2023): we initialize the final linear layer of DiT with zeros and otherwise we use the initialization techniques from the ViT (Dosovitskiy et al., 2020). We optimize our models using AdamW (Kingma & Ba, 2016; Loshchilov et al., 2017) with a constant learning rate $1e-4$, a weight decay 0.01, $(\beta_1, \beta_2) = (0.9, 0.999)$. For Purrception, we use eps $= 1e-6$. We also use a global batch size 256. Based on the training details of prior image generation methods, we compute the exponential moving average (EMA) of the backbone parameters over training using a decay rate of 0.9999, and we do inference using solely the EMA model.

Additionally, we use two tokenizers: (1) the Stable Diffusion's tokenizer `vq-f8` with a downsampling factor $f = 8$ and a codebook $\mathcal{C}$ of shape $16,384 \times 4$ (Esser et al., 2021) and (2) the LlamaGen's tokenizer `vq-ds8-c2i` with a downsampling factor $f = 8$ and a codebook $\mathcal{C}$ of shape $16,384 \times 8$ (Sun et al., 2024). This means that for a given RGB image $x$ of $256 \times 256$ resolution, the shape of the latent $z = \mathcal{E}(x)$ is $32 \times 32 \times d$ ($d = 4$ for `vq-f8` and $d = 8$ for `vq-ds8-c2i`), which is further quantized according to $\mathcal{C}$. During sampling, we use the decoder $\mathcal{G}$ to map the generated latent back into pixel space. The encoder, decoder, and codebook are kept frozen during training.

**Sampling and FID score computation.** Flow models need to simulate an ODE to solve the generative modeling task. We use the `torchdiffeq` library in PyTorch and (unless otherwise specified) the usual Euler method with 100 steps when generating samples.

For computing the FID scores, we first generate 10,000 samples for computing FID-10k scores and 50,000 samples for the FID-50k. Then, we use the `torch-fidelity` PyTorch library (Obukhov et al., 2020) to compute the FID score. For both FID-10k and FID-50k, we use 50k real samples (i.e., the entire validation set with $256 \times 256$ resolution) to compute the statistics for the target dataset.

Unless otherwise specified, we do *not* use classifier-free guidance for the models trained conditionally on ImageNet-1k.

**Computational resources.** All methods were trained using a distributed setup on the LUMI supercomputer, utilizing a total of 16 AMD MI250x GPUs, each equipped with 128GB of HBM2e memory.

## D   TRAINING STABILITY

Architecturally, we use a diffusion transformer (DiT) Peebles & Xie (2023) as a backbone to predict the codebook indices that compound the target datapoint. One of the biggest challenges we encountered was maintaining a stable training for larger DiT variants (i.e., DiT-L/2, DiT-XL/2), especially because such training instabilities occurred in the later stages of the training phase.

Since the major difference between training Flow Matching and Purrception is the training objective (i.e., mean-squared error for the former one, cross-entropy for the latter one) and Flow Matching does not have such training instabilities, we hypothesize that the cause might be the final softmax operation. Indeed, this divergence in the output logits from the log probabilities has been reported often as an instability issue by the research community when training large models at scale (Chowdhery et al., 2023; Wortsman et al., 2023). In their paper, Wortsman et al. (2023) name this issue the *logit drift problem*. To mitigate this issue, they propose regularizing the training using an additional *z-loss* which proved effective in training recent state-of-the-art, billion-parameter models such as Chameleon (Chameleon Team, 2024).

Inspired by its success, we apply z-loss regularization as well. Similar to Chameleon Team (2024), we add $10^{-5} \log^2 Z$ to Purrception's loss function, where $Z = \sum_{i=1}^{K} e^{x_i}$ and $\{x_i\}_{i=1}^{K}$ are the logits outputted by the backbone. Figure 7 shows Purrception achieves stability when z-loss is integrated. Thus, we used the z-loss by default when training Purrception.

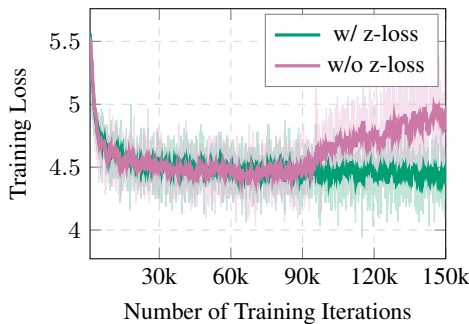

Figure 7: **Training loss curves with and without z-loss.** An additional z-loss avoids training divergence. Raw data is shown in lighter colors, while exponentially smoothed curves (EMA) are shown in bold. We used the same hyperparameters for both runs and a DiT-XL/2 backbone. EMA smoothing factor is $\alpha = 0.9$.

## E   THE EFFECT OF VQ AUTOENCODERS

Table 2: **Quantitative comparison of Purrception trained with different VQ tokenizers.** Evaluation on ImageNet-1k $256 \times 256$ shows that the choice of tokenizer significantly influences generation quality, with `vq-ds8-c2i` outperforming `vq-f8` across all FID thresholds.

|  | vq-f8 | vq-ds8-c2i |
|---|---|---|
| rFID | 1.19 | 0.59 |
| FID ($\tau = 0.7$) | 11.80 | 7.44 |
| FID ($\tau = 0.8$) | 10.85 | 6.46 |
| FID ($\tau = 0.9$) | 10.82 | 7.03 |
| FID ($\tau = 1.0$) | 12.33 | 9.60 |

When we train Purrception, we train exclusively the DiT backbone while keeping the encoder, decoder, and codebook vectors frozen. To test how much the performance of Purrception relies

on the VQ tokenizer, we train two DiT-XL/2 models on ImageNet-1k $256 \times 256$ with the same training configurations, for 3.5M iterations each, using two different VQ autoencoders: (1) the Stable Diffusion's `vq-f8` tokenizer (Rombach et al., 2022) and (2) the LlamaGen's `vq-ds8-c2i` tokenizer (Sun et al., 2024).

The results in Table 2 show that Purrception's performance is tightly coupled to the quality and design of the underlying VQ autoencoder used to produce the latent tokens. We can observe that for LlamaGen's tokenizer (which has a better rFID score as compared to Stable Diffusion's one), we obtain better FID-50k scores in class-conditional ImageNet-1k $256 \times 256$. This indicates that even when the DiT backbone is trained identically, the representational granularity and perceptual fidelity of the VQ tokenizer have a decisive impact on downstream generation quality.

Overall, the experiment highlights that the performance of Purrception is not tokenizer-agnostic: its effectiveness depends on the inductive biases and compression characteristics of the VQ autoencoder used. Future work could explore training custom VQ models co-optimized with the DiT backbone or investigating hybrid tokenizers that balance perceptual fidelity and codebook compactness.

