# OpenReview forum: "Purrception: Variational Flow Matching for Vector-Quantized Image Generation"
_ICLR.cc/2026/Conference — ICLR 2026 Poster_

### Official Review · Reviewer_8vtk · 2025-10-31

**Soundness:** 3
**Presentation:** 2
**Contribution:** 3
**Rating:** 4
**Confidence:** 2

**Summary:**

The paper investigated flow-matching models for vector-quantized image generation. It pointed out the strengths and weaknesses of existing approaches: continuous flow models preserved geometric information but could not exploit categorical learning signals, while discrete models aligned with the quantized structure and allowed temperature-scaling control but failed to understand geometry. It then proposed Purrception, a variational flow-matching technique for vector-quantized image generation that combined the advantages of continuous and discrete flow models. Purrception learned categorical posteriors over codebook indices, which could be used for computing velocity fields in the continuous embedding space. Experiments showed that Purrception outperformed both continuous and discrete flow-matching baselines in both convergence speed and image generation quality on the ImageNet-1k 256 x 256 dataset.

**Strengths:**

- The paper discussed the strengths and weaknesses of existing flow-matching approaches for vector-quantized image generation. It then proposed to employ a variational flow-matching technique to combine the advantages of mentioned approaches.
- The paper is technically sound.
- Experiments showed that Purrception outperformed both continuous and discrete flow-matching baselines in both convergence speed and image generation quality on the ImageNet-1k 256 x 256 dataset.

**Weaknesses:**

- The middle side of Equation (13) seems to be incorrect.
- Section 4.1: Missing information on the averge NFE and the used temperature (I guess it is 0.9 based on Table 1).
- Section 4.2: From my understanding, this experiment only varies the temperature at inference while keeping the training temperature (possibly 0.9) unchanged.
  + As expected, using the inference temperature similar to the training one (the straight-forward configuration) provides the best performance. Deviating the inference temperature from the training one causes performance degradation, particularly in color saturation. Hence, the experiment cannot prove the benefits of the temperature-scaling control
  + A more reasonable experiment is to use the same temperature for training & inference, and compare models trained w/ different temperatures
  + There is no evidence showing that the temperature affects image diversity
- The results reported in Table 1 are far from state-of-the-art for Class-conditional generation on ImageNet-1k 256 x 256. The state-of-the-art ones, e.g., REPA, have FID less than 2. The reported result is only strong for vector-quantized image generation using standard DiT backbones. The authors should correct their claim.
- Writting issues:
  + Equation 12: $\pi$ was used before being defined
  + L265: The text should be in the same paragraph with the previous one
  + Figure 5: Should add the temperature lable for each column

**Questions:**

- The middle side of Equation (13) seems to be incorrect.
- Section 4.1: Missing information on the averge NFE and the used temperature (I guess it is 0.9 based on Table 1).
- Section 4.2: From my understanding, this experiment only varies the temperature at inference while keeping the training temperature (possibly 0.9) unchanged.
  + As expected, using the inference temperature similar to the training one (the straight-forward configuration) provides the best performance. Deviating the inference temperature from the training one causes performance degradation, particularly in color saturation. Hence, the experiment cannot prove the benefits of the temperature-scaling control
  + A more reasonable experiment is to use the same temperature for training & inference, and compare models trained w/ different temperatures
  + There is no evidence showing that the temperature affects image diversity

---

> ### Author Response · Authors · 2025-11-21
>
> We would like to thank you for the feedback. The weaknesses and questions you raised helped us improve significantly the way we present our work, particularly Section 4.2. We answer each question / weakness one-by-one:
>
> **Weaknesses 2 & 3:** During training, **we use the default softmax temperature of 1.0, and only change temperature during inference**. Contrary to what would be expected, the best softmax temperature during sampling (i.e., that achieves the best FID) is not 1.0, because a value of 0.8 or 0.9 leads to better FID scores. *The qualitative and quantitative results we highlight in this section actually prove the benefits of the temperature-scaling control (which require no additional training cost).* **We rewrote the entire Section 4.2 to enhance readability and clarity**.
>
> Moreover, we already included all details about sampling in Appendix C in the paper. For your question, we used an ODE solver with 100 integration steps for all methods.
>
> **Weakness 4:** Indeed, the results reported in Table 1 are not quite on par with SOTA for the ImageNet-1k 256x256 class-conditional generation task, and contain various hard to compare methods (different tokenizers, different number of training iterations, different models sizes, etc). The only truly ‘direct’ comparison we did was between CFM, DFM, and Purrception, and showed improvement w.r.t both baselines. However, based on comments made by reviewers woJ8 and BoRK, we experimented with a better tokenizer and achieved better FID score compared to our current reported result. We also rewrote Section 4.3.
>
> **Weaknesses 1 & 5:** We checked and corrected all the writing issues mentioned (mathematical notations / inconsistencies, adding softmax temperature used in Figure 5).
>
> Thank you once again! We are happy to bring additional clarifications (if needed).

---

> ### Author Response · Authors · 2025-11-27
>
> Dear Reviewer 8vtk,
>
> Thank you once again for taking the time and effort to review our submission. As the deadline for the discussion phase is approaching, we wanted to kindly check if you have any additional questions or concerns that we could address in our rebuttal.
>
> We appreciate your insights and look forward to your feedback.
>
> Best regards,
>
> The Authors

---

> > ### Comment · Reviewer_8vtk · 2025-11-28
> >
> > Thanks to the authors for the rebuttal. It addressed my concerns pre-rebuttal. I recommend highlighting the modified text using a different text color to enhance readability.
> >
> > I have checked other updates and have some questions:
> > - None of the FID scores of Purrception in Table 2 matches the one in Table 1. Are the numbers in Table 2 for unconditional image generation, leading to the mismatch? If yes, why were different settings used for experiments in Tables 1 and 2?
> > - The authors claimed to add some preliminary results of SiT training in response to Reviewer i3Vq within a few days. I would love to see those results.

---

> > > ### Author Response · Authors · 2025-12-02
> > >
> > > Thanks for the questions and we are happy the rebuttal addressed your concerns. We uploaded again the paper marking with red all changes we made for the rebuttal. Below you can find the answers to your questions:
> > >
> > > - In both Table 1 and 2 we do **class-conditional generation**. The difference between Table 1 and Table 2 is that in Table 2 we do ***not*** use any classifer-free guidance (cfg). The goal of Table 2 is to show that the choice of the tokenizer influences generation quality, whereas in Table 1 we aimed for a low FID score and hence we used classifier-free guidance (which is well-known for its improving visual quality of samples).
> > > - The results are already in the rebuttal version. There are several variants of SiT depending on the type of interpolant, sampler, or time learning used. In Section 4.1 (Convergence speed), we chose SiT with deterministic sampler and continuous time learning, which *is essentially CFM* where we predict the endpoint via MSE. We motivate this choice better in Section 4.1.
> > >
> > > We hope this answers your follow-up questions!

---

### Official Review · Reviewer_woJ8 · 2025-11-01

**Soundness:** 1
**Presentation:** 3
**Contribution:** 2
**Rating:** 4
**Confidence:** 5

**Summary:**

This paper proposes variational flow matching to train flow matching with discrete code from vq-vae. By jointly combining discrete and continuous, it shows that flow matching achieves better convergence than purely continuous or discrete.

**Strengths:**

1. The paper shows that under same discrete VQ-VAE, it outperforms the discrete flow matching training or the continuous flow matching training. The idea of uncertainty in sampling is quite interesting.

2. The paper writing is easy to follow and clear.

**Weaknesses:**

1. The motivation is not well-convincing to me. It is not clear why receiving categorical signal is better for continuous flow matching. What is the geometry structure in continuous here ?

2. The theory about VFM seems to be out of place for me. To my understanding, this method is based on VFM framework but instead of inputing the codebook index and let the flow matching learn internal codebook embedding, they utilize the codebook embedding from vq-vae to create soft embedding $z_1$ and input continuous signal. This leads to limited novelty.

3. The model underperforms with DiT using continuous VAE. Furthermore, this technique heavily depends on a good VQ-VAE which is limited the model performance. It would be better to choose other discrete VQ-VAE, which has better rFID and some generative model training on that achieves very good FID like [1,2] to see how good the proposed technique can reached given different VQ-VAE.

[1]: An Image is Worth 32 Tokens for Reconstruction and Generation
[2]: Autoregressive Model Beats Diffusion: Llama for Scalable Image Generation

**Questions:**

Please see the weakness above

---

> ### Author Response · Authors · 2025-11-21
>
> We would like to thank you for the feedback. The weaknesses you pointed out are valid and helped us improve the quality of the paper. Below we reply to each of them separately in more detail:
>
> **Weaknesses 1 & 2:**
>
> **VQ latent spaces are a discrete set of points in a continuous (geometric) latent space**. The benefit of Purrception is that we have a *continuous flow in latent space*, but make sure during training *the model only attends to these **finite (categorical) number of points***. Hence, rather than being just geometric/continuous (FM) without this finite-point bias, **or** just discrete without leveraging the geometry of the encoder (e.g., DFM), Purrception sits in the middle of both methods.
>
> **The novelty of our method is exactly this hybrid formulation, which outside of diffusion for language modeling has been explored as far as we are aware**. Besides showing strong performance in general, our work is in line with a recent surge in ‘hybrid’ language models, and we show that through using VQ latent spaces we can leverage these hybrid benefits outside of the language domain. ***Finally, all of the above is formally justified through the VFM framework, which also gives theoretical support for Purrception.***
>
> **Weakness 3:**
>
> **To validate the effect of the tokenizers, we reran Purrception with a better tokenizer and observed significantly improved performance**. We listened to your suggestion and we used the vq_ds8_c2i tokenizer (with a codebook of shape 16,384 x 8) released by [2]. We currently have an FID score of 4.03 (~15% lower than our initial result) *with only half the number of iterations compared to the previous setup*.
>
> Indeed, the model doesn't quite reach SOTA FID in the setup we evaluated under. However, we do show that a hybrid approach outperforms and/or is competitive against several SOTA methods (continuous / discrete) with similar model size. Note that the methods in Table 1 contain many different approaches (auto-regressive, diffusions, flows) trained with different tokenizers/encoders, for different number of training steps. What we provide is evidence that _(1) Purrception is on par with such models (Table 1), and (2) using the same encoder/decoder and number of iterations ourperforms its continuous and discrete counterparts (Section 4.1)_.
>
> Thank you once again! We are happy to bring additional clarifications (if needed).

---

> ### Author Response · Authors · 2025-11-27
>
> Dear Reviewer woJ8,
>
> Thank you once again for taking the time and effort to review our submission. As the deadline for the discussion phase is approaching, we wanted to kindly check if you have any additional questions or concerns that we could address in our rebuttal.
>
> We appreciate your insights and look forward to your feedback.
>
> Best regards,
>
> The Authors

---

### Official Review · Reviewer_BoRK · 2025-11-03

**Soundness:** 2
**Presentation:** 3
**Contribution:** 3
**Rating:** 2
**Confidence:** 2

**Summary:**

This paper introduces Purrception to address a dual nature of VQ-latents, which the authors argue that existing flow-based models make a poor trade-off: Continuous Flow Matching (CFM) respects the continuous geometry but ignores the discrete categorical structure, while Discrete Flow Matching (DFM) models the indices but discards the geometry.

Purrception proposes a hybrid solution by adapting Variational Flow Matching (VFM). The model learns a categorical variational posterior over the discrete codebook indices, while the actual transport velocity is computed in the continuous embedding space.

Experimental results on ImageNet-1k $256\times256$ showing that Purrception converges significantly faster (1.7x-3.5x) and achieves a better final FID score than both CFM and DFM baselines when using the same DiT backbone88.

**Strengths:**

1. The motivation "discrete-continuous tradeoff" is a real problem for VQ-latents, and the paper's solution directly addresses it without compromise.
1. The proposed solution is simple and elegant.
1. Exprimental results show faster convergence than other DFM and CFM baselines.

**Weaknesses:**

1. Purrception failed to beat LlamaGen-XL, which has similar number of parameters, with respect to FID (Purrception 4.72 vs LlamaGen-XL 3.39), as well as other baselines.
1. Purrception's encoder, decoder, and codebook are kept frozen during training, thus the entire method is fundamentally capped by the quality of the frozen vq-f8 tokenizer.
1. Missing comparision with CDCD mentioned in related work, which also "preserves the continuous-time formulation by operating on noisy embeddings while training with cross-entropy".
1. Table 1 does not include a CFM baseline. The paper only argues the converge curve "strongly suggests" Purrception would surpass CFM, but this is not convicing enough.
1. Missing visual comparison with baselines.

**Questions:**

1. As shown in Figure 5, the temperature does not affect the generated image too much. How is temperature useful in real application? And any tests on DFM baseline?
1. Is z-loss just a specific patch that masks a deeper problem with scaling Purrception?
1. Any reason why adapting VFM to a fixed, pretrained codebook is novel compared to the CDCD framework?
1. Any evident that the final FID of 4.72 is not a property of Purrception, but just an inherent limitation of the vq-f8 latent space itself?
1. Did you test how Purrception performs with any other VQ-VAE tokenizers?

---

> ### Author Response · Authors · 2025-11-21
>
> We would like to thank you for the feedback. The weaknesses and questions you pointed out helped us improve the paper quality and strenghten our results. Below you can see the answers for each of your questions:
>
> **Weakness 2 & Questions 4-5 (Tokenizer):** Because we keep the encoder, decoder, and codebook frozen during training, a major point you addressed is that the relatively poor quality of the tokenizer clouds the performance results of Purrception. **To this end we tested as recommended by Reviewer woJ8 with a tokenizer that has a better rFID on ImageNet-1k 256x256 [1].** By using the vq_ds8_c2i tokenizer (with a codebook of shape 16,384 x 8), we currently have an FID score of 4.03 (~15% lower) *with only half the number of iterations compared to the previous setup (Stable Diffusion’s vq-f8 tokenizer)*. We hypothesize this will be further improved if it’s trained for longer time.
>
> **Question 2 (Z-Loss):** This phenomenon is not a patch in our method specifically. The *logit drift problem* is a common, well-researched issue met when training large models at scale caused by the final softmax operation. The additional z-loss helped us train Purrception for 3.5M iterations without reporting training instabilities for a rather large number of codebook vectors K = 16.384.
>
> **Question 1 (Softmax temperature):** Indeed, most qualitative differences are related to rather small details rather than significant changes. *In real applications, it can be used as an additional parameter that controls the level of details / image quality.* But maybe more importantly is that using the default softmax temperature of 1.0 during inference leads *qualitatively* *and quantitatively* to worse samples. While during training we used a constant temperature of 1.0, this does not mean the best softmax temperature is 1.0: we showed that a temperature of 0.8 - 0.9 achieves higher-quality samples, for instance. *Reviewer 8vtk suggested us to mention which temperature we used in Figure 5: this should highlight the visual differences better.*
>
> For DFM, we can in principle apply temperature scaling to the logits, but because it commits to hard index selections at each step, adjusting the temperature has little practical effect. The samplig collapses to discrete jumps regardless of the distribution’s softness. **We will highlight this point better in Section 4.2.**
>
> **Weakness 4 (Purrception vs. CFM comparison):** We fully agree with this point. Currently, we trained CFM for 2M iterations and compared CFM vs. Purrception for up to 2M training iterations instead of 1M. We showed that the gap is even larger now: **for DiT-XL/2 backbone, Purrception reaches the FID score of CFM trained on 2M iterations 2.3x faster (vs. 1.7x as pointed out after 1M training iterations).** We are currently training it for 3.5M iterations and show the final FID for CFM in Table 1.
>
> **Weakness 1 (Purrception vs. other SOTA baselines):** Table 1 does not show an “apple-to-apple” comparison. Note that the methods in this table contains many different approaches (auto-regressive, diffusions, flows) trained with different tokenizers/encoders, for different number of training steps, and having different models sizes. What we provide is evidence that (1) Purrception is on par with such models (Table 1), and (2) using the same encoder/decoder and number of iterations ourperforms its continuous and discrete counterparts (Section 4.1).
>
> **Weakness 3 & Question 3 (Comparison with CDCD):** Yes, CDCD is indeed closely related in spirit, though it is limited to diffusion and primarily developed for language embeddings rather than VQ latents. Our formulation is slightly more general: we propose a hybrid discrete–continuous relaxation that can be applied to flows in VQ spaces, but could in principle also be used for diffusions or other generative families. In that sense, CDCD can be viewed as a special instantiation of this broader hybrid view. We fully agree that the formalisms are close; the key differences are that our work brings this hybrid perspective to flow matching, introduces a principled inference interpretation, and adds components such as temperature-controlled posteriors that, to our knowledge, are new and enable learning hybrid flows in a stable and scalable way.
>
> Thank you once again! We are happy to bring additional clarifications (if needed).
>
> [1]: Autoregressive Model Beats Diffusion: Llama for Scalable Image Generation

---

> > ### Comment · Reviewer_BoRK · 2025-11-24
> >
> > Thanks for your clarification. Most of my concerns are addressed. I am going to raise my score. Please update the submission with new results and clarification.
> >
> > Besides, I still have one more question. As you reported,
> >
> > > To this end we tested as recommended by Reviewer woJ8 with a tokenizer that has a better rFID on ImageNet-1k 256x256 [1]. By using the vq_ds8_c2i tokenizer (with a codebook of shape 16,384 x 8), we currently have an FID score of 4.03 (~15% lower) with only half the number of iterations compared to the previous setup (Stable Diffusion’s vq-f8 tokenizer).
> >
> > using a different tokenizer (codebook) led to better results. Can the tokenizer and codebook be trained (or fine-tuned) together with Purrception? And can it lead to better results?

---

> > > ### Author Response · Authors · 2025-11-24
> > >
> > > Thanks for the great question. This is something we explored during development, and we think it’s a very promising direction for future work.
> > >
> > > In short: yes, the encoder, decoder, and codebook can be trained or adapted together with Purrception. We see two ways of doing this, 1) (somewhat trivially) through training everything jointly, and 2) through light-weight adapting the codebook.
> > >
> > > The straightforward way is to simply train from scratch the encoder, decoder, codebook, and the DiT backbone for Purrception jointly. However, this is quite expensive and outside our compute budget for this paper. But it’s certainly a valid and natural approach, and we expect it would improve results.
> > >
> > > Another option is to keep the encoder and decoder frozen but allow Purrception to slightly adjust the geometry of the codebook. One can do this by training a tiny adapter that predicts small offsets for each codebook vector. The model then uses these adapted positions when predicting the categorical distribution over codes, but we still decode using the original codebook so everything remains compatible. This gives you the best of both worlds: Purrception can reshape the codebook geometry in a flow-friendly way, while the pre-trained encoder/decoder remain untouched (i.e., the added compute is minimal).
> > >
> > > We ultimately didn’t explore this further because it adds an extra mechanism that distracts from the core message of the paper. However, we believe it’s a strong direction for follow-up research.
> > >
> > > Hope this helps!

---

> > > > ### Comment · Reviewer_BoRK · 2025-11-24
> > > >
> > > > Thank you. I have raised my score. Good luck :)

---

### Official Review · Reviewer_i3Vq · 2025-11-04

**Soundness:** 3
**Presentation:** 3
**Contribution:** 3
**Rating:** 4
**Confidence:** 3

**Summary:**

The authors propose a latent variational flow matching approach, Purrception, for vector quantized generative models. The motivation for the approach is to propose a hybrid between discrete and continuous latent flows for the latent generative models. They exploit the variational formulation of the marginal vector field and propose to model the discrete latent tokens as a categorical distribution (probabilities) from the continuous latent embeddings. With the experiments on the ImageNet-256 dataset, they show training efficiency over discrete and continuous latent generative models and improved final accuracy over discrete latent generative models.

**Strengths:**

The idea of a continuous flow path and categorical token prediction combined under the variational flow matching framework is interesting. The application to VQ-VAEs does show improvement in generative performance compared to latent discrete flow-based approaches.

**Weaknesses:**

Major:

Important baseline missing - The main improvement compared to baselines is shown in the training efficiency. However, a comparison to the key baseline latent generative model SiT (Ma et al. 2024) is missing. I would request the authors to add this to their training efficiency analysis to improve thoroughness.

Other:

The model doesn't reach competitive FID vs continuous models. This is not a major weakness, as it is known (empirically) that continuous latent models tend to perform better than the discrete ones in image generation. However, I would request the authors to add a discussion of why a 750M Purrception (discrete) will be more useful than a 675M DiT (continuous) model.

**Questions:**

In line 294, what is e^{x_i}?

---

> ### Author Response · Authors · 2025-11-21
>
> We would like to thank you for the feedback. Your review helped us provide a more thorough analysis when discussing the results and have a better literature review and comparison by including Scalable Interpolant Transformers (SiT). For every weakness and question, we offer a more detailed answer below:
>
> **We included SiT in the paper.** We agree with the major weakness pointed out. Right now, we implicitly made two changes when designing Purrception (as compared to CFM): (1) going from velocity prediction to endpoint prediction, and (2) going from MSE to cross entropy loss. We agree these should be ablated separately, and are running SiT (i.e., endpoint prediction with MSE) now. We aim to add some preliminary results in a few days. Moreover, we (1) add a paragraph about SiT and how it related to Purrception in the Related Work section, and (2) add SiT in Table 1 as a representative baseline.
>
> **What $e^{x_i}$ in line 294 is.** For a given i from 1 to K, where K is the number of codebook vectors in VQ-GAN, $x_i$ is the logit for the codebook vector i that has been outputted by the DiT backbone. $e^{x_i}$ is one of the K terms that give the sum Z. The goal is to calculate a regularization term named *z-loss*, whose purpose is to solve the much-researched *logit drift problem*. We also included some references in the Training Stability section.
>
> **750M Purrception (discrete) vs. 675M DiT (continuous) model.** Our motivation is that hybrid discrete-continuous architectures are becoming central across generative modeling, both in hybrid LLMs and discrete/continuous diffusion models (e.g., CANDI [1], CDCD [2]). These approaches suggest that combining continuous dynamics with discrete representational structure improves efficiency and robustness. Flows currently lack a principled hybrid formulation. This is the gap our method addresses, since VQ compression captures semantic units (textures, parts) that flows can exploit. The variational posterior enables flow learning without sacrificing this structure, serving as the bridge between discrete codes and continuous transport. In short: our hybrid VQ-variational formulation brings flow models into the emerging family of hybrid generative architectures, providing both a practical acceleration and a missing conceptual bridge between discrete latent modeling and continuous flow-based transport. We believe there is much to still explore in this direction, and will highlight this motivation stronger in the paper.
>
> Thank you once again! We are happy to bring additional clarifications (if needed).
>
> [1]: CANDI: Hybrid Discrete-Continuous Diffusion Models
>
> [2]: Continuous diffusion for categorical data

---

> > ### Author Response · Authors · 2025-11-27
> >
> > Dear Reviewer i3Vq,
> >
> > Thank you once again for taking the time and effort to review our submission. As the deadline for the discussion phase is approaching, we wanted to kindly check if you have any additional questions or concerns that we could address in our rebuttal.
> >
> > We appreciate your insights and look forward to your feedback.
> >
> > Best regards,
> >
> > The Authors

---

### Author Response · Authors · 2025-11-21
**Thank you to all reviewers and updates on the second version of the paper**

We would like to thank you very much for all the constructive feedback we received from all reviewers. The questions and weaknesses you pointed out helped us improve the paper a lot!

We addressed every comment raised by each reviewer separately below. However, we expect to upload a second version of the paper in the next few days, where we aim to include *mostly* the followings:

- improving the positioning of our work in literature
- validating Purrception's performance with better VQ tokenizers
- comparing Purrception with SiT baseline
- rewriting Section 4.2 to enhance clarity about softmax temperature scaling
- fixing small mathematical / writing inconsistencies

---

### Author Response · Authors · 2025-11-24
**Rebuttal version uploaded**

We would like to thank you very much once again for all the great feedback we received. We just uploaded the second version of our paper and your suggestions helped a lot with the way we present our method and results. Below you can find a short summary of what we added / changed:

- **We rewrote the entire Section 4 and we tested Purrception using LlamaGen’s tokenizer**. For all experiments, we made a more thorough analysis on the results we got:
    - **Section 4.3 (Quantitative results)**
        - Most reviewers complained about the bad FID score of 4.72 we got. Based on **BoRK and woJ8** suggestions, ***we trained Purrception on LlamaGen’s tokenizer and obtain a much lower FID score of 3.88. What’s more, we included in the Appendix E a better analysis of the performance Purrception obtains when trained with these two VQ tokenizers.***
        - We also make a more thorough analysis on the result we got, highlighting that Purrception is particularly competitive in the context of VQ image generation.
        - We added SiT as a continuous diffusion baseline in Table 1.
    - **Section 4.1 (Convergence speed)**
        - At the request of **i3Vq**, we added a new baseline inspired by the Scalable Interpolant Transformers (SiT) (Ma et al. 2024). To keep a fair comparison, we used the ODE version of SiT, that we named “CFM-endpoint” throughout the paper. *In the end, CFM-endpoint and CFM converge to a similar FID-10k score after 2M training iterations, being outperformed by Purrception for both DiT backbone variants.*
        - **BoRK** said that the previous plots “strongly suggest Purrception would surpass CFM” (i.e., if we let CFM train for longer). *In the current version, we train CFM for more training iterations and show that Purrception still leads to a lower FID in fewer iterations than CFM.*
    - **Section 4.2 (Softmax temperature scaling).** Based on the questions asked by **8vtk**, we rewrote the entire section to enhance clarity and readability. We also added in Figure 5 the softmax temperature used for every image.
- **We added Appendix E:** To find an answer of **BoRK’s question “**Any evidence that the final FID of 4.72 is not a property of Purrception, but just an inherent limitation of the vq-f8 latent space itself?**”,** we checked how well Purrception performs using Stable Diffusion’s and LlamaGen’s tokenizers. The results suggest that indeed the final FID of 4.72 is ***not*** a property of Purrception and that the performance depends on the tokenizer used.
- **We include few words about different parameterizations of flow matching in Section 5 (Related Work).**
- **We moved Section 3.3 (Training Stability) in Appendix D**. Due to the lack of space, we decided to move this section in the Appendix.
- **We corrected small mathematical notation inconsistencies highlighted by 8vtk in Section 3.**

---

### Meta-Review · Area_Chair_ZvhR · 2026-01-02

**Summary:**

This paper proposes a variational flow matching approach for VQ-based image generation. It attempts to address the discrete–continuous trade-off in VQ latents. The key idea is to learn categorical posteriors over codebooks while computing velocity fields in the continuous embedding space, combining the strengths of continuous flow matching (geometry-aware transport) and discrete models (categorical supervision). Reviewers’ concerns focused on whether the training efficiency gains were supported by the right baselines (notably SiT), how clearly the work is positioned relative to closely related approaches such as CDCD, and whether the reported ImageNet-1k 256×256 results are fundamentally limited by the quality of the frozen VQ tokenizer.

Post-rebuttal, the paper presents a technically sound and conceptually clean hybrid formulation with clear advantages over continuous and discrete flow-matching baselines.

**Reviewer Concerns:**

- Several major concerns were addressed in the rebuttal. The missing SiT baseline highlighted by i3Vq was acknowledged as a valid weakness; the authors clarified the two key design changes (endpoint prediction vs. velocity prediction and cross-entropy vs. MSE) and added SiT-related comparisons and discussion.
- Tokenizer dependence, raised by multiple reviewers, was directly studied by evaluating a stronger tokenizer, leading to improved FID and supporting the authors’ claim that performance is dependent on the VQ autoencoder.
- Concerns about z-loss were resolved by linking it to the logit drift problem, and questions about softmax temperature were clarified by showing that inference-time temperature scaling improves quality without retraining
- Some concerns remain partially outstanding, notably the close conceptual relationship to CDCD and the broader generalization beyond ImageNet-256, but these are appropriately framed as limitations and future work.

**Reviewer Scores:**

i3Vq (4) is likely to move to weak accept given the added SiT analysis.
BoRK (2) explicitly stated that concerns were addressed and that they raised their score; likely borderline accept / accept.
woJ8 (4) Core empirical concerns addressed; likely to remain borderline.
8vtk (4) stated that rebuttal addressed concerns; likely accept.

Based on these, I recommend acceptance of the paper as a poster.

---

### Decision · Program_Chairs · 2026-01-26

Accept (Poster)